# Site-Specific Nutrient Diagnosis of Orange Groves

**Danilo Ricardo Yamane** [1,*], **Serge-Étienne Parent** [2], **William Natale** [3], **Arthur Bernardes Cecílio Filho** [1], **Danilo Eduardo Rozane** [4], **Rodrigo Hiyoshi Dalmazzo Nowaki** [1], **Dirceu de Mattos Junior** [5] **and Léon Etienne Parent** [2,6]

1 Department of Plant Production, São Paulo State University (UNESP), Jaboticabal 14884-900, SP, Brazil
2 Department of Soils and Agri-Food Engineering, Université Laval, Québec, QC G1V 0A6, Canada
3 Department of Plant Science, Federal University of Ceará, Fortaleza 60020-181, CE, Brazil
4 Department of Agronomy, São Paulo State University (UNESP), Registro 11900-000, SP, Brazil
5 Instituto Agronômico de Campinas (IAC)/Centro de Citricultura Sylvio Moreira, Cordeirópolis 13490-000, SP, Brazil
6 Department of Soils, Federal University of Santa Maria, Santa Maria 97105-900, RS, Brazil
* Correspondence: danilo_yamane@yahoo.com.br

**Abstract:** Nutrient diagnosis of orange (*Citrus sinensis*) groves in Brazil relies on regional information from a limited number of studies transferred to other environments under the ceteris paribus assumption. Interpretation methods are based on crude nutrient compositions that are intrinsically biased by genetics X environment interactions. Our objective was to develop accurate and unbiased nutrient diagnosis of orange groves combining machine learning (ML) and compositional methods. Fruit yield and foliar nutrients were quantified in 551 rainfed 7–15-year-old orange groves of 'Hamlin', 'Valência', and 'Pêra' in the state of São Paulo, Brazil. The data set was further documented using soil classification, soil tests, and meteorological indices. Tissue compositions were log-ratio transformed to account for nutrient interactions. Ionomes differed among scions. Regression ML models showed evidence of overfitting. Binary ML classification models showed acceptable values of areas under the curve (>0.7). Regional standards delineating the multivariate elliptical hyperspace depended on the yield cutoff. A shapeless *blob* hyperspace was delineated using the *k*-nearest successful neighbors that showed comparable features and reported realistic yield goals. Regionally derived and site-specific reference compositions may lead to differential interpretation. Large-size and diversified data sets must be collected to inform ML models along the learning curve, tackle model overfitting, and evaluate the merit of *blob*-scale diagnosis.

**Keywords:** centered log ratio; machine learning; nutrient balance; local diagnosis

## 1. Introduction

Brazil is the world leader in orange production with a total area of 682.167 ha, of which 63% is located in the state of São Paulo [1]. The successful combinations of scions and rootstocks, balanced fertilization, and pest management form the first rampart against productivity loss in Brazilian orange orchards [2–5]. Nevertheless, the productivity of Brazilian orange orchards remains below crop potential due in part to imbalanced fertilization [6–8]. Balanced crop nutrition may in turn reduce pest problems [9].

Compared to surface soil testing, tissue testing is more suitable to guide fertilization because trees have access to nutrients deeper in the soil [10]. [11]. However, ionomes may differ among scions and rootstocks [3,12,13]. There is a growing interest in Brazil to update tissue nutrient references at the grove scale where genetic, managerial, and environmental growth factors vary widely [14,15].

Tissue analytical data are commonly interpreted using 'critical' nutrient concentration ranges that neglect nutrient interrelationships [16]. Indeed, plant nutrition is regulated by a network of genetically controlled physiological processes that impact tissue compositions [17]. Multiple levels of nutrient interactions result not only in synergistic or

antagonistic effects [18], but also evolve from chemical similarities between elements that are networking and self-adjusting to each other within the tissue compositional space [19]. Tissue compositions are unique assemblages of intrinsically multivariate data in which components cannot be interpreted in isolation [20–23]. Proximate [24] and remote sensing [25] studies showed that the interpretation of spectra to diagnose nutrient stresses should consider several nutrients simultaneously. There is thus a need to diagnose plant nutrition as full compositional networks [19,26].

Nutrient interactions have been reported traditionally as dual ratios [27]. Diagnosis and Recommendation Integrated System (DRIS) dual-ratio standards [27] have been elaborated for orange orchards in the USA [28], Venezuela [29], and Brazil [12,30]. The DRIS standards elaborated so far in Brazil were based on a small number of groves [30]. Nevertheless, DRIS has conceptual flaws. Dual-ratio standards are biased by including false positive specimens (cases of luxury consumption, suboptimal concentration, etc.). The DRIS variables are neither reflective nor additive despite attempts to correct such defaults [28,31]. The DRIS also promotes universality and timelessness [32], which were proved to be wrong [33–35]. Distortions in the DRIS were addressed using log-ratio transformations [20]. Log-ratios such as centered log-ratios (*clr*) project the constrained compositional data (e.g., between 0 and 100%, 1000 g kg$^{-1}$, or $10^6$ mg kg$^{-1}$) into the real space ($\pm\infty$) to allow the conducting of a multivariate analysis in the Euclidian space [36]. Log-ratios are useful in discriminating the ionomes of fruit species [23] and scions [4], as well as in ranking nutrients in the order of their limitation to yield to support fertilization decisions [15].

The distribution of tissue nutrient data has been traditionally determined using a multinormal distribution represented by ellipsoids with a centroid and variance [23]. However, successful agroecosystems may be unevenly distributed in the compositional hyperspace, and some of them may even be located outside the ellipses [37]. As a result of the complexity of agroecosystems and inherent interactions, the tissue nutrient space of nutritionally balanced and high-yielding specimens may show shapeless distribution patterns such as 'islands' or '*blobs*' [14,37,38]. A '*blob*' is a collection of nutritionally balanced and high-yielding groves that show features comparable to those of the diagnosed grove but are the limiting ones. The '*blob*' concept reflects a grower's propension to make comparisons with successful neighbors [11].

Successful agroecosystems can be detected using machine learning regression or classification models. Machine learning (ML) is a general term representing a wide variety of models used to process data sets and make predictions [39]. The ML can incorporate numerical and categorical variables [40,41]. The ML methods require much fewer assumptions than the traditional statistical methods [42]. The prediction ability of ML models is retrained by overfitting [43]. The ML models are increasingly applied to make predictions in biological and agronomic sciences [14,15,34,44–46] such as soil classification and mapping [47,48], carbon sequestration [49], image analysis [50,51], disease diagnosis [52], and the prediction of crop yields [53–55].

We hypothesized that: (1) the yield of orange groves could be predicted accurately by ML models; and (2) compositional nutrient diagnoses would be similar at regional and *blob* scales. The objective of this paper was to develop site-specific diagnostic tools to elevate low fruit yields to the locally documented yield potential.

## 2. Material and Methods

### 2.1. Experimental Setup

The data set comprised 551 observational data surveyed from 2012 to 2014 in rainfed orange orchards across the Central South region of the state of São Paulo, Brazil. Sites were located between −23.5812 and −21.7764 LAT and −49.5197 and −48.0364 LONG. The altitude of the orchards varied from 488.55 to 718.87 m. The 7–15-year-old groves belonged to the young-age group (>6 years of age) in which tree nutrition and production are stabilized [12,56]. There were two rootstocks ('Citrumelo Swingle' and 'Tangerina

Sunki') and three scions ('Hamlin', 'Valência', and 'Pêra') (Table 1). Because 'Hamlin' and 'Valência' were grafted uniquely onto 'Citrumelo Swingle' and 'Pêra' onto 'Tangerina Sunki', the rootstock effect could not be tested. The flowering period did not vary among the scion X rootstock combinations (Table 1). 'Hamlin' X 'Citrumelo Swingle' showed the earliest harvesting period, followed by the intermediate 'Pêra' X 'Tangerina Sunki' and the late 'Valência' X 'Citrumelo Swingle'. Fruit yields were reported as kg tree$^{-1}$ or tons ha$^{-1}$. Tree planting density varied from 220 to 830 plants ha$^{-1}$. The plot area averaged 15 ha.

**Table 1.** Number of observations, yield ranges, and flowering and harvesting periods for surveyed 'Hamlin' X 'Citrumelo Swingle', 'Valência' X 'Citrumelo Swingle', and 'Pêra' X 'Tangerina Sunki'.

| | 'Hamlin' | 'Pêra' | 'Valência' |
|---|---|---|---|
| No. of observations | 121 | 126 | 300 |
| | | tons ha$^{-1}$ | |
| Minimum yield | 18.8 | 6.4 | 1.1 |
| Median yield | 62.8 | 39.4 | 52.1 |
| Maximum yield | 136.4 | 102.4 | 141.4 |
| | | 2012 season | |
| Flowering period | September–October 2012 | September–October 2012 | September–October 2012 |
| Harvest period | May–June 2013 | July–October 2013 | October–December 2013 |
| | | 2013 season | |
| Flowering period | September–October 2013 | September–October 2013 | September–October 2013 |
| Harvest period | May–June 2014 | July–October 2014 | October–December 2014 |

The climatic regime of the region is Aw according to the Köppen–Geiger classification. The climate is tropical with a dry winter season and the rainfall concentrated in the summer season. The prevailing soils are Oxisols (Ustox) and Ultisols (Udults and Ustults) [57]. Oxisols (lateritic soils) and Ultisols (tropical Podzols) are altered soils high in Fe and Al oxi-hydroxides and contain kaolin clay. There were 205 Red Oxisols, 210 Red-Yellow Oxisols, and 136 Red-Yellow Ultisols in the data set. 'Hamlin' X 'Citrumelo Swingle' was grown on 48 Red Oxisols, 43 Red-Yellow Oxisols, and 30 Red-Yellow Ultisols. 'Pêra' X 'Tangerina Sunki' was grown on 26 Red Oxisols, 45 Red-Yellow Oxisols, and 55 Red-Yellow Ultisols. 'Valência' X 'Citrumelo Swingle' was grown on 131 Red Oxisols, 122 Red-Yellow Oxisols, and 51 Red-Yellow Ultisols. Soils may show hard setting layers at 20+ cm [58,59], but the depth to hardpan was not documented. Both soil groups are naturally acidic low-nutrient soils that require liming and fertilization [60].

### 2.2. Crop Management

Mineral fertilization [12] is intended to offset crop nutrient removal and loss and meet the nutrient demand for fruit development and new growth of leaves, branches, and roots [4]. In Brazil, the average nutrient offtake per tons of fresh fruit was estimated at 1.2 kg of nitrogen (N), 0.18 kg of phosphorus (P), 1.54 kg of potassium (K), 0.57 kg of calcium (Ca), 0.12 kg of magnesium (Mg), 0.09 kg of sulfur (S), 1.6 g of boron (B), 0.39 g of copper (Cu), 2.1 g of iron (Fe), 0.38 g of manganese (Mn), and 0.40 g of zinc (Zn) [61]. Based on a soil chemical analysis and the expected yield, the P and K application rates were 0–160 kg P ha$^{-1}$ and 0–200 kg K ha$^{-1}$, respectively [4]. The N rates varied from 70 to 240 kg N ha$^{-1}$ based on the leaf N concentration and expected yield. Fertilizers were split-applied on three occasions during the spring and summer. The B, Zn, and Mn were supplied as foliar sprays in the spring and summer [4]. Some orchards received 2 kg B ha$^{-1}$ year$^{-1}$ as a soil-applied fertilizer. Copper was mainly supplied through fungicide applications or metallic copper. The objective of liming was to reach 70% base saturation in the 0–20 cm layer during a long period [60]. Gypsum was also indicated to tackle Al in the 20–40 cm layer where Al saturation of CEC exceeds 40% and to supply calcium. Other management practices were carried out as recommended [61].

### 2.3. Meteorological Data

The daily heat units (DHU) were computed as follows [62]:

$$DHU \ (^{\circ}C) = T2M - 13^{\circ}C \tag{1}$$

where $T2M$ is the daily average daily temperature at a 2 m height and 13 °C is the minimum temperature for growth. The $T2M$ was reported via satellite using Boltzman's law to transform the surface radiation into temperature followed by spatialization ($0.5^{\circ} \times 0.65^{\circ}$). Because $DHU$ and $T2M$ were perfectly correlated, we retained $T2M$ as a meteorological index. Precipitations were recorded on daily basis then cumulated per month [59]. Where temperatures exceeded 30–35 °C and the rainfall and relative air humidity were low, flowers and fruits may have dropped but were not recorded. The most critical period extended from flowering (September/October) to a fruit diameter of up to 50 mm (end of December).

### 2.4. Tissue Analysis

Twenty-five trees were randomly selected for leaf sampling in each plot. Four mature (six-month-old) leaves were collected per tree from fruit-bearing shoots (3rd or 4th leaf from fruit) when the fruit size was 2–4 cm in diameter [4]. In total, 100 leaf samples were composited in each commercial grove. Leaves were gently washed successively with distilled water, a detergent solution (0.1%), a solution with hydrochloric acid (0.3%), and deionized water to reduce the surface contamination by dust and fungicides. The samples were oven-dried at 65 °C for 48–96 h and ground to less than 2 mm. The nitrogen (N) was analyzed using the micro-Kjeldahl method. Phosphorus (P), potassium (K), calcium (Ca), magnesium (Mg), sulfur (S), boron (B), copper (Cu), zinc (Zn), manganese (Mn), and iron (Fe) were quantified using ICP after acid digestion [63].

### 2.5. Soil Analysis

Soils were sampled in the 0–20 and 20–40 cm layers in each grove. Samples were composited, air-dried, and then ground and sieved to <2 mm. Soils were analyzed for pH (0.01 M CaCl$_2$), organic matter content, K, Ca, Mg, and (H + Al) [64]. The P was extracted using an exchange resin Amberlite IRA-400 (20–50 mesh), quantified via colorimetry using the ascorbic acid method, and reported as mg dm$^{-3}$. The K, Ca, and Mg were extracted using an exchange resin Amberlite IRA-120 (20–50 mesh), quantified via flame photometry (K) or atomic absorption spectrophotometry (Ca, Mg), and reported as mmol$c$ dm$^{-3}$. The potential acidity (H+Al) was derived from the SMP pH buffer methodology [65]. The cation exchange capacity (CEC) was computed as the sum of the cationic species (K, Ca, and Mg) and the exchangeable capacity. The CEC of Brazilian Oxisols may vary from 17 to 134 cmol$_c$ kg$^{-1}$ compared 7–88 cmol$_c$ kg$^{-1}$ for Brazilian Ultisols, indicating a large variation in clay and organic matter contents [66].

### 2.6. Centered Log-Ratio Transformation

The tissue compositional simplex comprised N, P, K, Ca, Mg, S, Cu, Zn, Mn, Fe, and B concentrations expressed using the same measurement unit (g kg$^{-1}$). The filling value ($F_v$) used to allow the back-transformation of the log-ratio-transformed data into familiar concentration values [4,23] was computed as follows:

$$F_v = 1,000,000 - \left( \text{sum of quantified nutrient concentrations reported in g kg}^{-1} \right) \tag{2}$$

The *clr* of each component ($clr_x$) was computed as the ratio of nutrient concentration of any component $xi$ to the geometric mean across the D components ($G$) to account for nutrient interactions as follows [20]

$$clr_{xi} = ln\frac{x}{G} \tag{3}$$

where the geometric mean (*G*) was computed as follows:

$$G = ([N] \times [P] \times [K] \times [Ca] \times [Mg] \times [S] \times [B] \times [Cu] \times [Fe] \times [Mn] \times [Zn] \times [Fv])^{\frac{1}{D}} \quad (4)$$

As a result, the *clr* variable integrated all pairwise ratios in the compositional simplex as follows:

$$clr_{xi} = ln\left(\frac{xi}{N \times P \times \ldots \times F_v}\right) = ln\left(\frac{xi}{N} \times \frac{xi}{P} \times \ldots \times \frac{xi}{F_v}\right) \quad (5)$$

The *clr* variables added up to zero.

*2.7. Statistical Analysis*

A discriminant analysis (DA) was conducted to compare the ionomes, the soil properties in upper (0–20 cm) and lower (20–40 cm) layers, and the soil classes. Tissue nutrient balances were transformed into orthonormal isometric log-ratios (*ilrs*) prior to the DA analysis [14,15]. Machine learning (ML) models were run using the R 'caret' package [67]. A supervised ML classification model approximated the function that predicts the outcome of interest based on the relationship between target variable and features or predictors [68]. The data matrix ($X_{ij}$; $x_i = [x_{i1}, x_{i2} \ldots, x_{ij}]$) was a sample composed of j predictors. The target variable was crop yield. The features were scion X rootstock combinations, plant age, tree density (for crop yields expressed as kg tree$^{-1}$), centered log-ratio-transformed tissue compositions (N, P, K, Ca, Mg, S, Fe, Mn, Cu, Zn, and B), soil classification, soil test results for the 0–20 and 20–40 cm layers (organic matter content, pH, P, K, Ca, Mg, and exchangeable acidity), and monthly meteorological indices (*T2M* and monthly precipitation). The clay content and cation exchange capacity (CEC) were excluded as features because they were derived from already-documented features (organic matter content and cationic species) and thus provided redundant information.

Since any ML algorithms can produce overfitting [69], a validation data set was required to evaluate the model. The data set was split into training (70%) and testing (30%) sets [67]. Data were preprocessed with 'zv' to identify zero-variance predictors, 'center' and 'scale' to provide a simple location and scale the transformation of each predictor, 'spatialSign' to project predictor values onto a unit circle, applying x* = x/||x||, and the 11 nearest neighbors (kmax = 11), a distance of two, and a kernel set at optimum. Statistical analyses were conducted in the R statistical environment [70]. Log-ratio transformations were computed using the R 'compositions' package [71]. Machine learning modeling was conducted using the R 'caret' package [67].

Features were selected for their contribution to model accuracy whatever their statistical significance [72]. The most accurate ML model was random forest. The ML models were first run as regression models. The model precision was measured as the R$^2$ value and as the root-mean-square error (RMSE). The target variables were the fruit yield expressed in kg tree$^{-1}$; plant density was added as feature. To run the binary random forest classification models, the target variable was fruit yield in tons per ha$^{-1}$; i.e., the product of fruit yield per tree and tree density per ha, as a familiar unit to select a yield goal by growers. Yield cutoff values for the binary classification were set at 50 or 60 tons ha$^{-1}$. The latter cutoff value is the one generally chosen by Brazilian growers. The 50-ton ha$^{-1}$ yield cutoff was tested to include 'Pêra' X 'Tangerina Sunki' specimens among high yielders at the regional scale. Model precision was measured as the area under the curve (AUC) and classification accuracy (CA). An AUC of 0.5 had no diagnostic interest [73]. The model was little informative in the AUC range of 0.5 to 0.7, moderately informative if $0.7 \leq AUC < 0.9$, and very informative if $AUC \geq 0.9$. Where the AUC of training and testing models were acceptable, data classification was further conducted in cross-validation after merging the training and testing data sets.

Observations were partitioned by the confusion matrix of the ML classification model into four groups as follows [23]: (1) true negative (TN) specimens (high performance for nutritionally balanced groves), which formed altogether the reference population; (2) true

positive (TP) specimens (low performance for nutritionally imbalanced groves); (3) false negative (FN) specimens (low performance for nutritionally balanced groves (type II error); and (4) false positive (FP) specimens (high performance for nutritionally imbalanced groves (type I error). Accuracy (Acc), the proportion of specimens correctly classified as balanced or imbalanced [23], was computed as follows:

$$[Acc = [(TN + TP)/(TN + FN + TP + FP)]] \tag{6}$$

The negative predictive value (NPV) was the probability that nutrient balance returned a high grove performance. The positive predictive value (PPV) was the probability that the nutrient imbalance returned a low grove performance. Specificity was the probability that a high grove performance was nutritionally balanced. Sensitivity was the probability that a low grove performance was nutritionally imbalanced. The NPV, PPV, specificity, and sensitivity were computed as follows [23]:

$$NPV = \frac{TN}{(TN + FN)} \tag{7}$$

$$NPV = \frac{TP}{(TP + FP)} \tag{8}$$

$$Specificity = \frac{TN}{(TN + FP)} \tag{9}$$

$$Sensitivity = \frac{TP}{(TP + FN)} \tag{10}$$

*2.8. Delineation of the Regional and Blob Spaces*

The regional multivariate space was the multinormal elliptical distribution of TN grove specimens set apart from the regional survey data set (Figure 1). Some TN specimens may have been located outside the swarm of TN specimens, thereby losing important information on successful groves. While the ML classification model returned a probability to reach the cutoff yield, there was no indication of an attainable yield goal at the grove scale. The shapeless *blob* multivariate space was delineated by local features [38]. The nutrient compositions of the diagnosed specimen were compared to those of TN specimens in the *blob* where most features were similar under the *ceteris paribus* assumption (Figure 1).

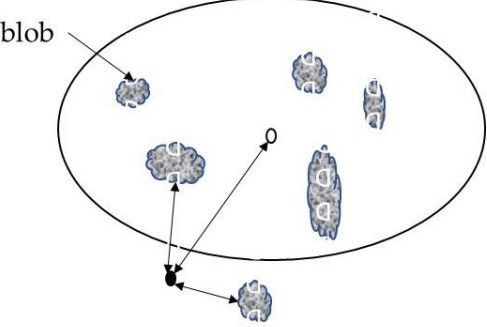

**Figure 1.** Elliptical distribution of true negative (TN) specimens under the classical assumption of multinormal distribution where nutrient centroids are represented by the empty circle. Data are distributed as 'Islands' or '*blobs*'. Some TN specimens are located in '*blobs*' outside the ellipse. The diagnosed specimen (dark circle) is located at shorter distance from two *blobs*' centroids compared to the regional centroids.

*2.9. Regional and 'Blob'-Scale Nutrient Standards*

Tissue nutrient standards were computed at the regional scale as the *clr* means ($clr_i^*$) and standard deviations ($SD_i^*$). The number of TN specimens depended on the yield cutoff

used to run the classification model. While the concentration values were not additive to allow computing contrasts between the compositions, the *clr*s of two equal-length compositions could be contrasted. The *clr*s could rank nutrient indices in the order of nutrient limitations. The $I_{clr_i}$ indices were computed as follows [20]:

$$I_{clr_i} = \frac{clr_i - clr_i^*}{SD_i^*} \tag{11}$$

where $clr_i$ is the ith *clr* value of the diagnosed specimen. Positive and negative $I_{clr_i}$ indices indicate relative nutrient excess and shortage, respectively, and can be reported in a histogram.

It is common for growers to compare abnormal to normal plants both growing in otherwise similar conditions [11]. We selected *k* TN specimens compositionally close to the diagnosed specimen to rank nutrients in the order of their limitation to yield. The *k*-closest successful TN neighbors formed a locally representative *blob*, which is a concept of agroecosystem similarity developed by [38] that has been successfully applied to agricultural crops [37,74,75]. Depending on the number of TN specimens in the *blob*, the *clr* indices ($I_{clr_i}^{blob}$) were computed using *blob clr* centroids as follows and reported in histograms:

$$I_{clr_i}^{blob} = clr_i - clr_i^{blob} \tag{12}$$

or, depending on the number of close neighbors:

$$I_{clr_i}^{blob} = \frac{clr_i - clr_i^{blob}}{SD_i^{blob}} \tag{13}$$

To delineate the *blob*, we selected *k* TN specimens that presented features close to those of the diagnosed specimen. Because the *clr* variables had a Euclidian geometry, a distance $\varepsilon$ between two equal-length compositions *a* and *b* could be computed as follows:

$$\varepsilon = \sqrt{\sum_{i=1}^{D} \left( clr_i^a - clr_i^b \right)^2} \tag{14}$$

The plant, meteorological, and soil features of the TN specimens were retrieved from the data set to further support the comparison between the diagnosed specimen and those of the TN *blob*. Equation (12) can also be reported as a 'perturbation' vector (*p*) of the nutrient concentration ratios between the diagnosed and reference compositions (*) as follows [15]:

$$p = X \ominus x = \left\{ \frac{N}{N^*}, \frac{P}{P^*}, \dots, \frac{F_v}{F_v^*} \right\}, \tag{15}$$

The perturbation index was zero-scaled as $\frac{X}{x} - 1$ to illustrate the relative deficiency, sufficiency, or excess [76]. The perturbation vector returned the same ranking as the *clr* difference between two equal-length compositions if the geometric means were exactly the same (i.e., $G_i = G_i^{blob}$) as shown below:

$$I_{clr_i}^{blob} = clr_i - clr_i^{blob} = ln\left( \frac{x_i/G_i}{x_i^{blob}/G_i^{blob}} \right) \approx ln\left( x_i/x_i^{blob} \right), \tag{16}$$

The assumption of geometric mean similarity may thus hold where $G \approx G_i^{blob}$.

## 3. Results

### 3.1. Results of Tissue and Soil Tests

The concentration quartiles of the surveyed groves showed some deviations from the current Brazilian concentration ranges (Table 2). An excess of Cu in foliar tissues of the surveyed groves was the result of disease management that used Cu-based fungicides.

Excessive K levels were related to high K requirements and K fertilization; these can be antagonistic to tissue Ca and Mg. There was a large variation in the foliar Fe for 'Hamlin' X 'Citrumelo Swingle' and 'Valência' X 'Citrumelo Swingle', which indicated differences in the soil properties.

**Table 2.** Tissue test results for 'Hamlin' X 'Citrumelo Swingle', 'Valência' X 'Citrumelo Swingle', and 'Pêra' X 'Tangerina Sunki' compared to current Brazilian concentration ranges [3,77].

| | 'Hamlin' X 'Citrumelo Swingle' | | | 'Valência' X 'Citrumelo Swingle' | | |
|---|---|---|---|---|---|---|
| | Minimum | Median | Maximum | Minimum | Median | Maximum |
| | g kg$^{-1}$ | | | g kg$^{-1}$ | | |
| N | 21.6 | 25.6 | 33.0 | 18.2 | 25.7 | 36.1 |
| P | 0.9 | 1.2 | 2.7 | 0.8 | 1.2 | 3.0 |
| K | 8.0 | 13.9 | 70.7 | 6.50 | 13.6 | 87.1 |
| Ca | 13.8 | 34.6 | 49.7 | 17.5 | 35.7 | 56.0 |
| Mg | 1.8 | 3.3 | 7.9 | 2.0 | 3.7 | 8.4 |
| S | 1.8 | 2.6 | 21.3 | 1.7 | 2.6 | 23.4 |
| B | 0.038 | 0.099 | 0.211 | 0.028 | 0.097 | 0.251 |
| Cu | 0.005 | 0.052 | 0.333 | 0.005 | 0.064 | 0.545 |
| Zn | 0.014 | 0.037 | 0.147 | 0.012 | 0.041 | 0.154 |
| Mn | 0.018 | 0.052 | 0.200 | 0.011 | 0.049 | 0.176 |
| Fe | 0.040 | 0.129 | 3.974 | 0.048 | 0.116 | 3.746 |
| | 'Pêra' X 'Tangerina Sunki' | | | Current Brazilian standards | | |
| | Minimum | Median | Maximum | Lower Bound | Centroid | Upper Bound |
| | g kg$^{-1}$ | | | g kg$^{-1}$ | | |
| N | 20.1 | 24.1 | 35.0 | 25 | 27.5 | 30 |
| P | 0.8 | 1.2 | 1.7 | 1.2 | 1.4 | 1.6 |
| K | 8.8 | 13.7 | 27.1 | 10 | 12.5 | 15 |
| Ca | 12.5 | 32.8 | 56.0 | 35 | 42.5 | 50 |
| Mg | 1.6 | 3.1 | 5.3 | 3.5 | 4.2 | 5.0 |
| S | 1.6 | 2.5 | 3.2 | 2.0 | 2.5 | 3.0 |
| B | 0.036 | 0.065 | 0.201 | 0.050 | 0.100 | 0.150 |
| Cu | 0.007 | 0.063 | 0.486 | 0.010 | 0.015 | 0.020 |
| Zn | 0.015 | 0.043 | 0.218 | 0.035 | 0.053 | 0.070 |
| Mn | 0.019 | 0.045 | 0.165 | 0.030 | 0.045 | 0.060 |
| Fe | 0.041 | 0.107 | 0.292 | 0.050 | 0.010 | 0.150 |

The soil properties are presented in Table 3. There were apparently large differences in nutrient and lime management among growers. The P and K centroids were at low to medium levels according to Brazilian guidelines [78]. Soils were generally acidic with pH values ranging from 3.8 to 7.0 in the 0–20 cm layer and from 3.8 to 6.9 in the 20–40 cm layer. The base saturation of CEC ranged from 8 to 96% in the 0–20 cm layer and from 17 to 94% in the 20–40 cm layer compared to the 70% recommended for fruit crops [60]. Organic matter contents varied from 8 to 64 g dm$^{-3}$ in the 0–20 cm layer and from 8 to 67 g dm$^{-3}$ in the 20–40 cm layer. Clay contents ranged from 17 to 344 g kg$^{-1}$ in the 0–20 cm layer and from 15 to 360 g kg$^{-1}$ in the 20–40 layer. The CEC values, which reflected the large variation in clay and organic matter contents, ranging from 24 to 296 mmol$_c$ dm$^{-3}$ in the 0–20 cm layer and from 21 to 204 mmol$_c$ dm$^{-3}$ in the 20–40 cm layer compared to 17 to 134 mmol$_c$ dm$^{-3}$

for Oxisols and 7–88 mmol$_c$ dm$^{-3}$ for Ultisols documented in Brazil [66]. The average *T2M* seasonally varied in the range of 15.5 to 27.4 °C. The monthly rainfall ranged from 0.3 to 446.3 mm. Highly variable features may explain the strong variations in fruit yields.

**Table 3.** Statistics of soil properties in the 0–20 and 20–40 cm layers.

| Scion | pH (CaCl$_2$) | SOM | P | K | Ca | Mg | (H+Al) † | CEC | Base Saturation |
|---|---|---|---|---|---|---|---|---|---|
| | | g dm$^{-3}$ | mg dm$^{-3}$ | | | mmol$_c$ dm$^{-3}$ | | | % |
| | | | | **0–20 cm layer** | | | | | |
| 'Hamlin' | 5.19 ± 0.55 | 25 ± 12 | 34 ± 25 | 3 ± 2 | 30 ± 26 | 14 ± 10 | 29 ± 14 | 75 ± 38 | 57 ± 16 |
| 'Valência' | 5.30 ± 0.54 | 22 ± 13 | 32 ± 20 | 2 ± 1 | 26 ± 17 | 12 ± 8 | 24 ± 12 | 63 ± 29 | 60 ± 15 |
| 'Pêra' | 5.06 ± 0.56 | 15 ± 6 | 28 ± 21 | 2 ± 1 | 19 ± 10 | 9 ± 5 | 21 ± 8 | 50 ± 17 | 56 ± 14 |
| | | | | **20–40 cm layer** | | | | | |
| 'Hamlin' | 5.06 ± 0.73 | 25 ± 15 | 32 ± 49 | 2 ± 1 | 28 ± 24 | 12 ± 9 | 26 ± 11 | 69 ± 35 | 57 ± 15 |
| 'Valência' | 5.05 ± 0.50 | 21 ± 13 | 24 ± 24 | 2 ± 1 | 24 ± 18 | 11 ± 8 | 23 ± 8 | 60 ± 29 | 57 ± 14 |
| 'Pêra' | 4.73 ± 0.46 | 14 ± 6 | 18 ± 16 | 2 ± 1 | 16 ± 11 | 8 ± 6 | 22 ± 7 | 47 ± 18 | 50 ± 15 |

† Exchangeable acidity.

The discriminant analysis showed that the centroids of small ellipses differed significantly ($p = 0.05$) among the scion X rootstock combinations and soil classes (Figure 2). As a result, the scion X rootstock combinations, soil classes, and soil properties in the 0–20 cm and 20–40 cm layers could be included as predictors that contributed to the outcome.

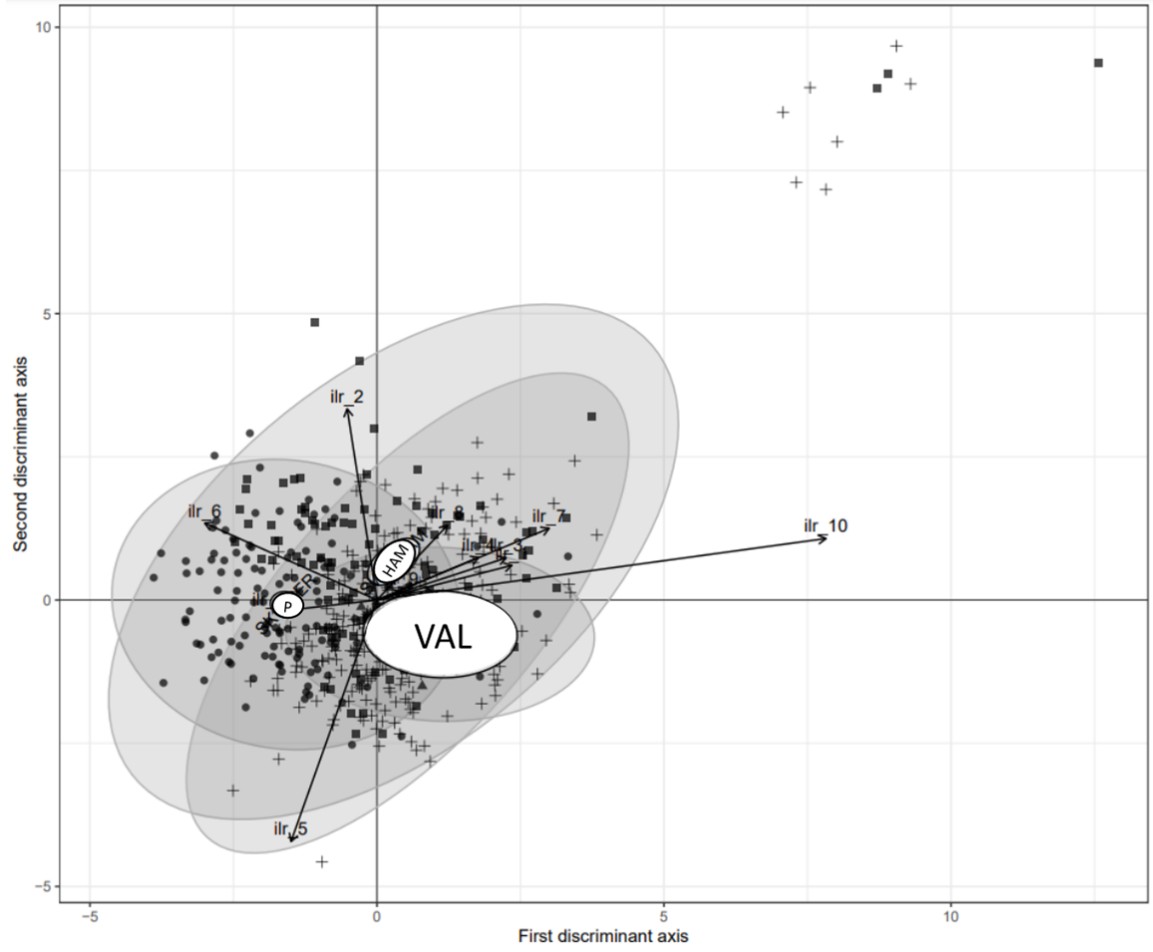

**Figure 2.** *Cont*.

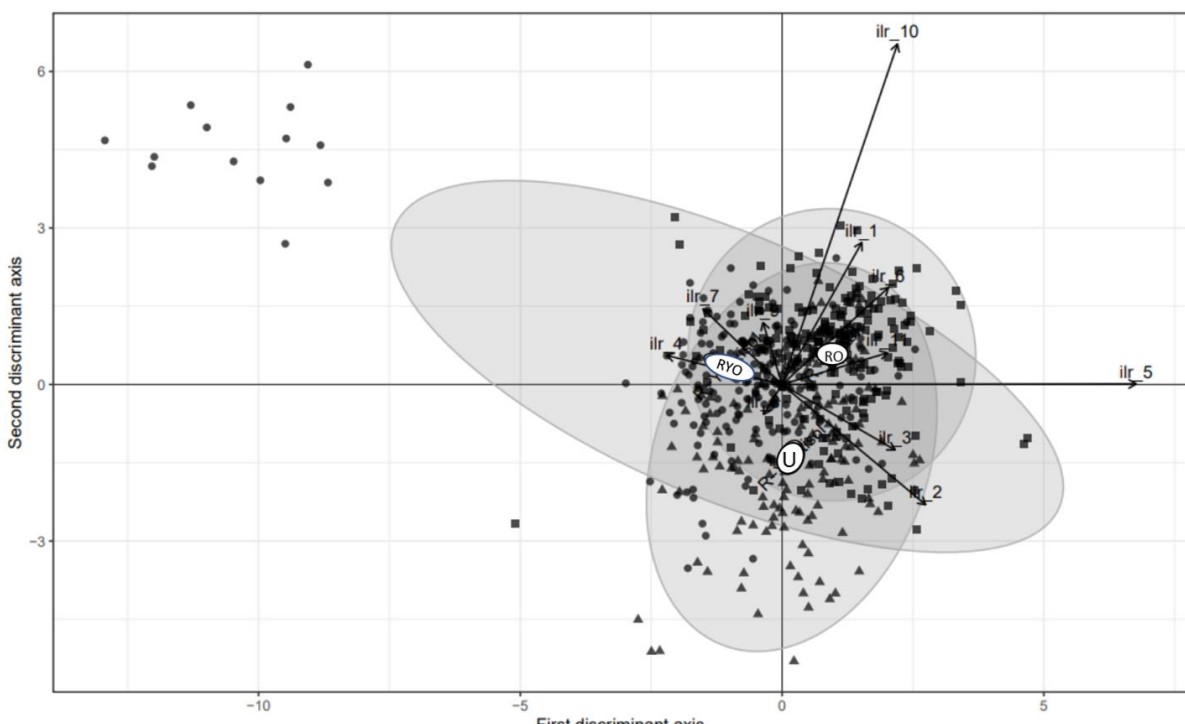

**Figure 2.** Discriminant analysis of scion ionomes (**upper** figure) and soil test results in the 0–20 cm and 20–40 cm layers of soil classes (**lower** figure). P = 'Pêra' X 'Tangerina Sunki' (●); HAM = 'Hamlin' X 'Citrumelo Swingle' (■); VAL = 'Valência' X 'Citrumelo Swingle' (+); RYO—Red-Yellow Oxisol (●); RO = Red Oxisol (■); U = Red-Yellow Ultisol (▲); *ilr* = orthonormal isometric log-ratio.

### 3.2. Random Forest Regression Models

The feature contribution to the regression model was tested by removing successively meteorological indices, soil tests in the 20–40 cm layer, soil tests in the 0–20 cm layer, and plant features (scion, age, and nutrient balances). The performance of the regression model showed a close fit for the training data set ($R^2$ = 0.860 to 0.905) but did not perform as well on the test data ($R^2$ = 0.086 to 0.520) (Table 4). The smaller error rates in the training compared to the testing showed evidence of overfitting due to noisy information learned and memorized under the training. Because the $R^2$ values increased and the RMSE decreased as more features were added to capture the complexity of the agroecosystems, the model's performance improved along the learning curve. Additional key features were likely needed to support the model but were not documented in the data set. Nonetheless, we tested whether the classification random forest models were still acceptable using in-hand features.

### 3.3. Random Forest Classification Models

The random forest binary classification models were validated using the test data set and via cross-validation. Model performance varied with the yield cutoff and feature selection (Table 5). The AUC exceeded 0.7 whatever the selected yield cutoff when at least scion, age, and nutrient balances were included as features. The soil type, meteorological indices, and soil tests contributed to a smaller extent than plant features (scion, age, and tissue tests). At yield cutoff of 50 tons ha$^{-1}$, plant features, and soil properties in the 0–20 cm layer sufficed to reach a high AUC value (0.801). A minimum data set would comprise the tissue test and soil test in the 0–20 cm layer, the scion and tree age as routinely acquired by growers, and the daily meteorological data and soil classification available from state records.

**Table 4.** Performance of the random forest regression model after partitioning the data set into training (70%) and testing (30%).

| Features | Yield as tons ha$^{-1}$ § | | Yield as kg Tree$^{-1}$ † | |
|---|---|---|---|---|
| | R$^2$ | RMSE | R$^2$ | RMSE |
| Training data set | | | | |
| Temperature, rainfall, scion, age, tissue nutrients, soil classification, S1, S2 | 0.905 | 7.295 | 0.913 | 14.544 |
| Scion, age, tissue nutrients, soil classification, S1, S2 | 0.898 | 7.571 | 0.907 | 15.061 |
| Scion, age, tissue nutrients, soil classification, S1 | 0.897 | 7.593 | 0.905 | 15.196 |
| Scion, age, tissue nutrients, soil classification | 0.898 | 7.575 | 0.908 | 14.970 |
| Scion, age, nutrient balances | 0.897 | 7.586 | 0.899 | 15.647 |
| Nutrient balances | 0.860 | 8.852 | 0.885 | 16.674 |
| Testing data set | | | | |
| Temperature, rainfall, scion, age, tissue nutrients, soil classification, S1, S2 | 0.285 | 17.874 | 0.506 | 36.804 |
| Scion, age, tissue nutrients, soil classification, S1, S2 | 0.321 | 17.415 | 0.515 | 35.691 |
| Scion, age, tissue nutrients, soil classification, S1 | 0.257 | 18.217 | 0.520 | 36.274 |
| Scion, age, tissue nutrients, soil classification | 0.109 | 19.951 | 0.489 | 37.438 |
| Scion, age, nutrient balances | 0.144 | 19.562 | 0.494 | 37.248 |
| Nutrient balances | 0.086 | 20.210 | 0.423 | 39.759 |

§ Computed as the product of tree density and yield per tree; † plant density added as feature in those models; S1, S2: soil properties in the 0–20 and 20–40 cm layer, respectively.

**Table 5.** Performance of the random forest binary classification model at yield cutoff values of 50 and 60 tons ha$^{-1}$ using cross-validation (10 folds).

| Features | 50 tons ha$^{-1}$ | | 60 tons ha$^{-1}$ | |
|---|---|---|---|---|
| | AUC | Accuracy | AUC | Accuracy |
| Temperature, rainfall, scion, age, tissue nutrients, soil classification, S1, S2 | 0.796 | 0.748 | 0.806 | 0.740 |
| Scion, age, tissue nutrients, soil classification, S1, S2 | 0.797 | 0.730 | 0.813 | 0.750 |
| Scion, age, tissue nutrients, soil classification, S1 | 0.811 | 0.742 | 0.801 | 0.755 |
| Scion, age, tissue nutrients, soil classification | 0.799 | 0.748 | 0.799 | 0.731 |
| Scion, age, nutrient balances | 0.783 | 0.735 | 0.783 | 0.728 |
| Nutrient balances | 0.683 | 0.662 | 0.658 | 0.633 |

The 121 'Hamlin' X 'Citrumelo Swingle', 126 'Pêra' X 'Tangerina Sunki', and 304 'Valência' X 'Citrumelo Swingle' were partitioned into four quadrants in the confusion matrix (Table 6). At a yield cutoff of 60 tons ha$^{-1}$, there were 150 TN specimens but no 'Pêra' X 'Tangerina Sunki'. At a yield cutoff of 50 tons ha$^{-1}$, there were 261 TN specimens, of which 21 were 'Pêra' X 'Tangerina Sunki'. The NPV and specificity were the highest among 'Hamlin' X 'Citrumelo Swingle' groves, which indicated that high fruit yields were closely associated with adequate nutrient balances. 'Pêra' X 'Tangerina Sunki' groves showed the highest sensitivity. Quartile nutrient concentration ranges for TN specimens were computed at the scion X rootstock level and a yield cutoff value of 50 tons ha$^{-1}$ (Table 7). Compared to the other scion X rootstock combinations, the B concentration range was lower in the foliar tissues of 'Pêra' X 'Tangerina Sunki'.

**Table 6.** Partitioning of the true negative (TN), false negative (FN), false positive (FP), and true positive (TP) results among scion X rootstock combinations in the confusion matrix of the binary random forest model that included all features.

| Scion X Rootstock | TN | FN | FP | TP | Total | NPV | PPV | Specificity | Sensitivity | Accuracy |
|---|---|---|---|---|---|---|---|---|---|---|
| | | | **60 tons ha$^{-1}$** | | | | | | | |
| 'Hamlin' X 'Citrumelo Swingle' | 67 | 12 | 19 | 23 | 121 | 0.85 | 0.55 | 0.78 | 0.66 | 0.74 |
| 'Pêra' X 'Tangerina Sunki' | 0 | 4 | 21 | 101 | 126 | 0.00 | 0.83 | 0.00 | 0.96 | 0.80 |
| 'Valência' X 'Citrumelo Swingle' | 83 | 43 | 45 | 133 | 304 | 0.66 | 0.75 | 0.65 | 0.76 | 0.71 |
| | 150 | 59 | 85 | 257 | 551 | 0.72 | 0.75 | 0.64 | 0.81 | 0.74 |
| | | | **50 tons ha$^{-1}$** | | | | | | | |
| 'Hamlin' X 'Citrumelo Swingle' | 87 | 10 | 12 | 12 | 121 | 0.90 | 0.50 | 0.88 | 0.55 | 0.82 |
| 'Pêra' X 'Tangerina Sunki'. | 21 | 17 | 21 | 67 | 126 | 0.55 | 0.76 | 0.50 | 0.80 | 0.70 |
| 'Valência' X 'Citrumelo Swingle' | 153 | 49 | 29 | 73 | 304 | 0.76 | 0.72 | 0.84 | 0.60 | 0.74 |
| | 261 | 76 | 62 | 152 | 551 | 0.77 | 0.71 | 0.81 | 0.67 | 0.75 |

**Table 7.** Lower and upper quartiles of true negative specimens of 'Hamlin' X 'Citrumelo Swingle', 'Pêra' X 'Tangerina Sunki', and 'Valência' X 'Citrumelo Swingle' at yield cutoff value of 50 tons ha$^{-1}$.

| Nutrients | 'Hamlin' X 'Citrumelo Swingle' | | 'Pêra' X 'Tangerina Sunki' | | 'Valência' X 'Citrumelo Swingle' | |
|---|---|---|---|---|---|---|
| | Lower Quartile | Upper Quartile | Lower Quartile | Upper Quartile | Lower Quartile | Upper Quartile |
| N | 24.3 | 27.4 | 22.6 | 25.9 | 24.3 | 26.8 |
| P | 1.1 | 1.4 | 1.0 | 1.3 | 1.1 | 1.3 |
| K | 11.8 | 16.9 | 12.3 | 15.5 | 11.7 | 15.4 |
| Ca | 30.8 | 40.4 | 25.8 | 39.2 | 31.9 | 40.9 |
| Mg | 2.8 | 3.8 | 2.6 | 3.5 | 3.2 | 4.0 |
| S | 2.4 | 2.9 | 2.3 | 2.8 | 2.4 | 2.8 |
| B | 0.081 | 0.118 | 0.054 | 0.080 | 0.076 | 0.127 |
| Cu | 0.025 | 0.080 | 0.027 | 0.062 | 0.032 | 0.087 |
| Zn | 0.025 | 0.056 | 0.028 | 0.048 | 0.029 | 0.056 |
| Mn | 0.037 | 0.074 | 0.030 | 0.062 | 0.037 | 0.068 |
| Fe | 0.093 | 0.144 | 0.075 | 0.121 | 0.096 | 0.140 |

*3.4. Nutrient Standards at Regional Scale*

Regional TN *clr* standards were computed for each scion X rootstock combination (Table 8). The centroids and variances differed among scions, which indicated that the three scions should be diagnosed separately; this confirmed the results of the discriminant analysis (Figure 2) and of the random forest binary classification model (Table 5). The *clr* values of tissue N, P, K, and Mg of the early maturing 'Pêra' X 'Tangerina Sunki' tended to be high compared to the tissue components of other scions at a yield cutoff of 50 tons ha$^{-1}$. In contrast, the Ca, B, Cu, and Fe levels appeared to be low.

*3.5. Order of Nutrient Limitations at the Regional and Blob Scales: Example*

The random forest binary classification model could predict the performance of a given grove from features documented in the data set. The model classified the yield as a probability of high or low under the *ceteris paribus* assumption. At the *blob* scale, comparable genetic, managerial, and environmental conditions could be set apart in the data set to delineate the referential *blob*s of the comparable TN specimens [38]. Thereafter, nutrients could be ranked against the *clr* nutrient benchmarks in the selected referential *blob*.

**Table 8.** Nutrient standards as *clr* mean and standard deviation (SD) of 'Hamlin' X 'Citrumelo Swingle', 'Pêra' X 'Tangerina Sunki', and 'Valência' X 'Citrumelo Swingle' TN specimens at a yield cutoff of 50 tons ha$^{-1}$.

| Nutrients | 'Hamlin' X 'Citrumelo Swingle' | | 'Pêra' X 'Tangerina Sunki' | | 'Valência' X 'Citrumelo Swingle' | |
|---|---|---|---|---|---|---|
| | *clr* Mean | *clr* SD | *clr* Mean | *clr* SD | *clr* Mean | *clr* SD |
| | 50 tons ha$^{-1}$ | | | | | |
| N | 2.866 | 0.209 | 2.923 | 0.220 | 2.789 | 0.155 |
| P | −0.142 | 0.239 | −0.116 | 0.236 | −0.272 | 0.198 |
| K | 2.261 | 0.326 | 2.354 | 0.211 | 2.157 | 0.323 |
| Ca | 3.153 | 0.247 | 3.040 | 0.278 | 3.141 | 0.171 |
| Mg | 0.812 | 0.241 | 0.863 | 0.255 | 0.833 | 0.216 |
| S | 0.607 | 0.281 | 0.591 | 0.163 | 0.558 | 0.320 |
| B | −2.538 | 0.370 | −2.728 | 0.262 | −2.540 | 0.314 |
| Cu | −3.355 | 0.430 | −3.544 | 0.309 | −3.404 | 0.370 |
| Zn | −3.674 | 0.875 | −3.330 | 0.765 | −3.328 | 0.668 |
| Mn | −3.732 | 0.414 | −3.660 | 0.255 | −3.554 | 0.438 |
| Fe | −2.678 | 0.335 | −2.924 | 0.408 | −2.744 | 0.366 |
| Fv | 6.365 | 0.134 | 6.532 | 0.211 | 6.421 | 0.199 |
| | 60 tons ha$^{-1}$ | | | | | |
| N | 2.889 | 0.213 | - | - | 2.780 | 0.140 |
| P | −0.116 | 0.226 | - | - | −0.322 | 0.161 |
| K | 2.261 | 0.332 | - | - | 2.093 | 0.247 |
| Ca | 3.166 | 0.261 | - | - | 3.160 | 0.143 |
| Mg | 0.827 | 0.245 | - | - | 0.827 | 0.191 |
| S | 0.636 | 0.299 | - | - | 0.490 | 0.114 |
| B | −2.524 | 0.388 | - | - | −2.595 | 0.212 |
| Cu | −3.321 | 0.434 | - | - | −3.385 | 0.436 |
| Zn | −3.787 | 0.846 | - | - | −3.233 | 0.481 |
| Mn | −3.764 | 0.416 | - | - | −3.456 | 0.467 |
| Fe | −2.697 | 0.364 | - | - | −2.732 | 0.396 |
| Fv | 6.431 | 0.212 | - | - | 6.372 | 0.112 |

Let us diagnose a defective (true positive) tissue specimen of 'Valência' X 'Citrumelo Swingle' (Table 9). The fruit yield of the diagnosed specimen was 20 tons ha$^{-1}$. Compared to the current Brazilian nutrient ranges, there was an apparent relative excess of Cu and a sufficiency of others. The excess Cu was indicative of difficulties in disease control. This would mean little or no change in the fertilization regime despite the low yield for a specimen classified as true positive (low-yielding, nutritionally imbalanced specimen).

The relative order of the nutrient limitations to the yield was computed at the *blob* scale using the *k*-closest TN compositional neighbor (*k* = 1) or the *clr* means and standard deviations of the six closest TN neighbors (*k* = 6). The six closest neighbors were selected by ranking the Euclidean distances of the tissue compositions between the diagnosed specimen and the TN specimens. The soil type of the defective specimen was Red-Yellow Ultisol. The meteorological indices were similar between the diagnosed and referential TN specimens. The soil pH, organic matter content, and extractable elements (P, K, Ca, and Mg) in the upper layer (0–20 cm) were also comparable except for one Red-Yellow Ultisol that showed a low pH (pH 4.3) compared to those of the diagnosed specimen (pH 5.8). The yield was 53 tons ha$^{-1}$ in the *k* = 1 *blob* and averaged 67 tons ha$^{-1}$ in the *k* = 6 *blob*, which indicated a large yield gap between the diagnosed grove and the successful neighboring groves.

**Table 9.** Nutrient diagnosis of a defective 'Valência' X 'Citrumelo Swingle' specimen.

| Component | Concentration | Brazilian Nutrient Ranges § | | Specimen | Regional † | |
|---|---|---|---|---|---|---|
| | | g kg⁻¹ | | *clr* | | |
| N | 31.0 | 25 | 30 | 3.024 | 2.789 | 0.155 |
| P | 1.3 | 1.2 | 1.6 | −0.188 | −0.272 | 0.198 |
| K | 13.0 | 10 | 15 | 2.156 | 2.157 | 0.323 |
| Ca | 36.5 | 35 | 50 | 3.186 | 3.141 | 0.171 |
| Mg | 4.2 | 3.5 | 5.0 | 1.019 | 0.833 | 0.216 |
| S | 2.3 | 2.0 | 3.0 | 0.435 | 0.558 | 0.320 |
| B | 0.134 | 0.050 | 0.150 | −2.619 | −2.540 | 0.314 |
| Cu | 0.056 | 0.010 | 0.020 | −3.950 | −3.404 | 0.370 |
| Zn | 0.035 | 0.035 | 0.070 | −3.287 | −3.328 | 0.668 |
| Mn | 0.029 | 0.030 | 0.060 | −3.760 | −3.554 | 0.438 |
| Fe | 0.110 | 0.050 | 0.150 | −2.421 | −2.744 | 0.366 |
| Filling value | - | - | - | 6.404 | 6.421 | 0.199 |

† Across features under the *ceteris paribus* assumption; § Quaggio et al. (2022) [77].

The regional diagnosis returned a relatively large N excess; some P, Mg, and B excess; and a large Mn shortage (Figure 3). The *blob* diagnosis (*k* = 6) returned a large Ca excess and Mn shortage; some B and Cu excess; and some P, K, and S shortage. The *blob* diagnosis (*k* = 1) returned relative excess of N, Ca, and Cu and a relative shortage of K and Mn. This differential diagnosis posed a new challenge that has not been reported before.

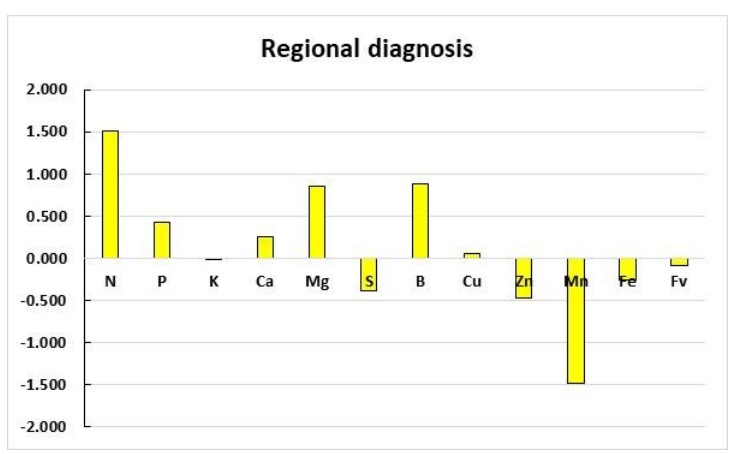

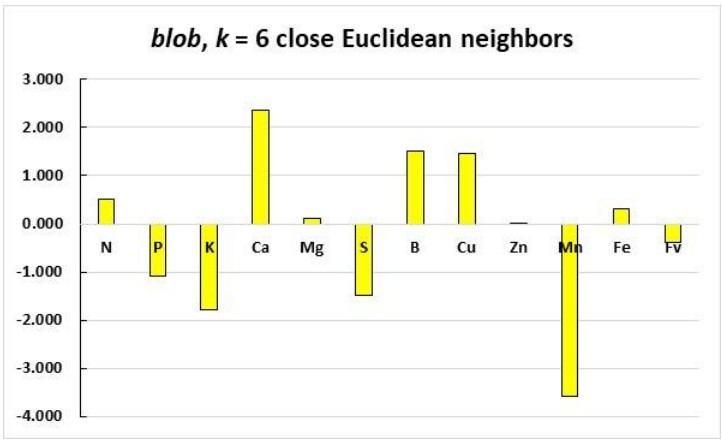

**Figure 3.** *Cont.*

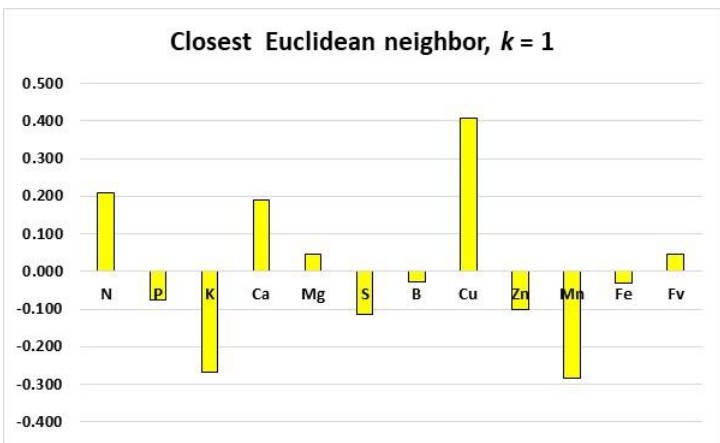

**Figure 3.** Order of nutrient limitations in a defective 'Valência' X 'Citrumelo Swingle' specimen diagnosed at regional or *blob* (*k* = 1 or 6) scales. The signs of *clr* indices indicate relative nutrient excess (+) or deficiency (−). Fruit yield averaged 53–67 tons ha$^{-1}$ in the *blobs* compared to 20 tons ha$^{-1}$ for the defective specimen.

## 4. Discussion

### 4.1. Contribution of Documented Features to ML Models

Orange yields in the surveyed region were impacted primarily by plant genetics, age, and nutrient balances (Tables 4 and 5). Soil properties in the 0–20 cm layer were also contributive. Indeed, the root system of the young orange trees was much more abundant in the 0–30 cm layer than below [59,79]. While features not yet documented such as the fertilization regime and disease control could likely have contributed to the yield, they were assumed to be applied as recommended. The fertilization regime and related features (placement, timing, source, and rate) could vary among growers. Where available, a grower's fertilization data could provide information on corrective measures to reach the yield potential at the minimum cost under comparable conditions. Scions could be grafted on different rootstocks depending on pedoclimatic conditions. Citrus rootstocks are selected for resistance to diseases and tolerance to drought, especially in hard-setting soils in which a cohesive soil mass located 20–70 cm below the surface limits water movement and root penetration [58,59]. The *blob* paradigm follows a grower's propension for comparisons with successful neighbors and Alexander von Humboldt's concept of natural systems that combines local features and knowledge [11,14]. The *blob* concept of local-scale diagnosis should be further tested via field experimentation and compared to current diagnostic methods.

The experimental data could validate the model prediction, measure the sensitivity of nutrient balances to nutrient additions, and provide response curves to support corrective measures [37,74,75]. Experimental data exist in various formats in Brazil [2,4,80–84], but data assemblage from different sources was beyond the scope of this study. Another issue was the carryover effects of carbohydrate and nutrient accumulations in the plant tissues of perennial plants that impact the following fertilization regime [85–88]. Several years of experimentation are needed to monitor the nutrition and fertilizer requirements of perennial crops given the information on carryover effects such as prior plant nutrient status, tree pruning, trunk diameter, and crop yield.

### 4.2. Nutrient Diagnosis

Differences between the nutrient concentration ranges assessed in the present study and others reported in Brazil [4] were attributable to: (1) TN specimens only in our reference group (excluding FP specimens), (2) tissue compositions analyzed as assemblages of nutrients rather than processed separately [34], (3) concentrations ranges tending to narrow down as more information on nutrient compositions and interactions was integrated [89], and (4) varietal effects. We compared the nutrient diagnoses using regional standards and

*blob* standards for the *k*-nearest neighbors. The *blob* provided realistic attainable yields under comparable growing conditions.

New questions arose on the relevance of current methods to diagnose the nutrient status of orange groves. Using *blob* or regional standards, the results of nutrient diagnosis may differ due to differences in centroids and supporting assumptions such as *ceteris paribus* (all factors but the ones being examined were assumed to be at equal or optimum levels), thus impacting the decision to adjust the fertilization regime to local conditions. Nutrient standards may also vary with the yield cutoff selected.

### 4.3. Boron Limitations

The B concentration ranges for 'Pêra' X 'Tangerina Sunki' differed compared to the other scions. This may have been due to genetics, managerial conditions, or environmental growing conditions. Boron deficiency is common in the tropics [89–91] because soil B availability is generally low [92]. Soil B is mobile in its predominantly nonionic form (boric acid) and may be leached through excessive rainfall [90]. Boron plays a central structural role in the formation of the xylem, root growth, and water transportation [93]. Boron concentrations in reproductive plant parts that contain higher levels of pectin, phenols, or sugar alcohol most often exceed the B concentrations detected in the vegetative parts. Boron is a hardener for plant cells and hence a protective mechanism against pests. Boric acid binds to sugars in cell walls and cross-links two chains of pectic polysaccharide through borate–diester bonding at the rhamnogalacturonan II region, which forms a network of polysaccharides [94]. In rainfed orange orchards, long drought periods affect root B transfer due to reduced mass flow [95]. Considering the low B mobility in the phloem of citrus [93], foliar fertilization is inefficient to fully meet the demand of new flushes of shoot growth and should be complemented by soil B application [61,90,96,97]. The B is commonly applied as a foliar spray [61,97].

### 4.4. Nutrient Excess and Shortage

The *clr* standards that incorporate all nutrients interacting in the plant tissue could also indicate overfertilization. There is a trend of overfertilizing orchards with N, P, and K because the macronutrient offtake through fruit harvest is substantial [61]. As a result, the fruit yield and quality may be affected by overfertilization driven by nutrient budgets rather than field experimentation and success stories documented in data sets [82–84]. N overfertilization may result in poor-quality fruits and a higher susceptibility to disease and insect feeding [9,82]. As is the case for other perennials [85–88], N requirements may vary widely in orange orchards [4] due in part to the carryover effects of carbohydrate and nutrient accumulations over time. Carbohydrate and nutrient reserves accumulated in off-years or previous years can be remobilized at a high rate in on-years [98]. The N stored in the tree biomass can be redistributed to developing leaves and fruits via the phloem [61]. Therefore, fruit production might not be impacted by a low nutrient supply during the current season. Nonetheless, if nutrient reserves are not replenished regularly, trees may undergo a gradual reduction in canopy density, which results in decreased fruit production in subsequent seasons [4]. This aspect could be addressed in field experiments and monitored via crop logging.

Due to strong sorption of phosphate ions by aluminum and iron oxyhydroxides in tropical soils such as Oxisols and Ultisols [99], nutrient-use efficiency of the fertilizer P is thought to be low [100–102] At low soil-test P values, citrus response to P fertilization was found to stimulate root–shoot growth [80] and increase the fruit yield [2,82]. The fruit yield can respond to P fertilization when the soil-test P is higher than 20 mg resin-P dm$^{-3}$ [2,82]. However, a yield loss can occur when the soil-test P exceeds 40 mg resin-P dm$^{-3}$ [81]. There is a high risk of P overfertilization [89]. The co-precipitation of P- and Zn-conducting vessels is exacerbated by an excess of P, which inhibits Zn transport from root to shoot and causes metabolic disorders [103]. While P fertilization has been guided traditionally by tissue diagnosis, soil analysis, and expected yield [4], a *blob*-scale diagnosis could provide

the realistic yield goals documented in the data set. The *blob* provides successful fertilization regimes under comparable conditions and can be used to derive response curves to added nutrients under comparable experimental conditions.

The K offtake due to a fruit harvest is substantial [61] and requires heavy K fertilization [82–84]. However, this may lead to a K excess in the soil and a relative K excess in foliar tissues, especially if the soil test is not taken into account or when K surpluses are large [61]. An excessive K dosage may lead to a yield loss due to K-Ca antagonism [83,104]. Ca accumulates even more than N in the tree biomass and correlates positively with fruit yield [105,106]. A soil pH correction through liming may not suffice to meet plant Ca demand. It has been recommended to complement Ca nutrition with gypsum [102] or calcium nitrate when plant Ca demand peaks [107]. Parsimonious K supply can contribute to re-establishing tissue K-Ca balance. To account for Ca, K, and Mg interactions in the soil, the relationship between the crop yield and soil test as exchangeable K, Ca, Mg, and acidity could be revisited using the nutrient-balance concept [108].

Relative Cu excess is common in the foliage of orange orchards due to Cu applications to tackle foliar diseases. Any Cu toxicity due to Cu accumulation may increase the oxidative stress and reduce plant growth (especially root growth), nutrient, water absorption, and photosynthesis rates [109,110]. While high soil-test Cu values are also commonly reported in orchards, other nutrients should be properly balanced to increase the resistance of orange trees to common diseases [6]. A concomitant S shortage may be problematic. S plays a central role in regulating cross-talk with cationic microelements [19]. Mn and Zn may be in short supply in Oxisols and Ultisols. Mn and Zn are sorbed strongly by soil colloids and are commonly at low levels in the parent material of tropical soils [3,61,111]. Mn and Zn are often supplied through foliar sprays [3,105,106]. Soil applications of those micronutrients are complementary to foliar sprays in sandy soils that are low in organic matter [111].

## 5. Conclusions

The random forest regression model that related fruit yields to meteorological, soil, and plant features showed evidence of overfitting. A random forest binary classification model was found to be acceptable based on the area under the curve. The scion X rootstock combination and grove age contributed more to the fruit yield than the meteorological and soil variables. The scion X rootstock combination, plant age, tissue nutrient balances, and soil properties in the 0–20 cm layer formed the minimum data set to reach an acceptable model AUC. The nutrient concentration ranges of the surveyed TN specimens at a regional scale depended on the yield cutoff.

There was a need to adjust the search for nutrient standards to yield a cutoff specific to the scion X rootstock combination and to focus on the local conditions delineated in the *blob* of the *k* neighboring successful agroecosystems. Using a defective 'Valência' X 'Citrumelo Swingle' TP specimen as an example, the regional and *blob* diagnoses differed. The predictions of the *blob* model, which were based Alexander von Humboldt's concept of natural systems that combines local features and knowledge and on grower's propension for comparisons with successful neighbors, should be further evaluated via field experimentation.

Citrus growers could adopt a concept of yield-limiting *clr* nutrient balances specific to their agroecosystems in which groups of nutrients are optimally balanced in referential *blobs*. Most nutrient standards currently used to adjust crop fertilization are derived from regional data sets with few features. More local key features such as those related to the soil profile, carryover effects, and pest and fertilizer management should be collected to build a diversified data set that enables an increase in the model's accuracy and improve the yield prediction. The *blob* paradigm based on Alexander von Humboldt's concept of natural systems that combines local features and knowledge and is deciphered using artificial intelligence tools [112] should be further compared to the current diagnostic methods via field experimentation.

**Author Contributions:** D.R.Y. collected and organized the data, elaborated the models, and cowrote the paper; S.-É.P. conceived, wrote, and calibrated models; W.N., A.B.C.F., D.E.R., R.H.D.N., and D.d.M.J. collected data and revised the paper; L.E.P. cowrote the paper. All authors have read and agreed to the published version of the manuscript.

**Funding:** The authors acknowledge the financial support of the Coordenação de Aperfeiçoamento de Pessoal de Nível Superior (CAPES) and the Natural Sciences and Engineering research Council of Canada #2254.

**Acknowledgments:** This project was elaborated under an interinstitutional agreement between Universidade Estadual Paulista (UNESP), Jaboticabal-SP, Brazil; and the Université Laval, Québec (QC), Canada.

**Conflicts of Interest:** The authors declare that this research was conducted in the absence of any commercial or financial relationships that could be construed as a potential conflict of interest.

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
