# Peer review of "Site-Specific Nutrient Diagnosis of Orange Groves"

_horticulturae, doi:10.3390/horticulturae8121126_

Round 1

Reviewer 1 Report

The study is quite novel and interesting. However there is a need to improve the methodology section.

1. Its is not clear if there were comparable treatments (independent vs dependent variables);

2. It is not clear as to how the soil sampling was done.

3. There is an aspect of crop management, but it is not clear how this was done or factored in the model

4. Laboratory analysis of the soils is not well described but only cited. It could be better explained in general terms (methods & equipment) but per parameter or group of parameters e.g exchangeable bases, trace-elements etc

Author Response

Dear Reviewers

We appreciate all the comments and suggestions provided. All were accepted.

As there were suggestions and complementary revisions among the reviewers, we have incorporated them all in a single file. All changes are highlighted.

Thanks again for the collaborations and we look forward to it.

Yours sincerely,

Prof. Dr. Danilo Eduardo Rozane

Reviewer 2 Report

Dear Authors,

Please adopt the proposed suggestion. 

Best regards

Author Response

Dear Reviewer

We appreciate all the comments and suggestions provided. All were accepted.

As there were suggestions and complementary revisions among the reviewers, we have incorporated them all in a single file. All changes are highlighted.

Thanks again for the collaborations and we look forward to it.

Yours sincerely,

Prof. Dr. Danilo Eduardo Rozane

Reviewer 3 Report

1. what "Mg" means in line26?

2.Line 50. what "DRIS" means? I think authors should referred the whole name when they used the short name first appearance. There are many similar missing in the whole manuscript. Please correct it all, even those commonly known short name like Cu, N, P are all need to indicate their whole name. 

3. why chose "acidity (H + Al)." to present acidity not ph or others? I think authors should provide some information in the introduction.

4.Does this "Where necessary, Ca and Mg were surface-applied as dolomitic limestone between April and June to reach 70% base saturation of the cation exchange capacity (CEC) and Mg concentration of at least 4 mmolc dm-3 in the 0-0.20 m layer " ,mean authors appied fertilizer to their  investigated sites? and which sites were used Ca and Mg? and this means when the built model is used in the future, we need to consider the condition of the soil first, and then to adjust the soil condition to a well level, and we can use the model finally? So the model is fited to nurtion or fit to a specific situaion?

 5.Line 120, what does the "plot" means here? and only "Four" leaves were analysised is a little analysis to present the nutrition condition for a whole tree. 

6.authors use fruit yield to represent the condition of the trees, however, the yield is not totally related to the Nutrient condition, it is reflected many different aspects form many different side. Authors hope to use the yield to be a index to evaluate Nutrient diagnosis will be a littel adventure.

7. fig1, authors used 6 canopy varieties in this research, why only chose 5 for foliar analysis and 3 for soil analysis?

Author Response

(The authors gave the same response as above.)

Round 2

Reviewer 3 Report

1. Although the authors add a lot of sampling information, we still don't know the actual situation. Since the source of the database is unclear, and this study is based on the database for AI modeling, it is difficult to assess the quality of this research.

The authors mention in the abstract that 716 orchards were sampled, but only 25 were sampled for soil (line 117). There is no way to tell how many orchards the leaves were taken from, we only know that 100 leaves were sampled from each orchard. Which cultivar of each orchard were planted for each sampled soils and leaves? Are they sufficient to represent the so-called 716 orchard plots? Judging from the current information, only 3% (25/716) of the data sources have analyzed soils, how to confirm the author's hypothesis"" We hypothesized that varietal ionomes and "environmental factors" can be amalgamated to generate regional nutrient standards across Brazilian orange orchards using tools of machine learning and compositional data analysis. "? The authors specifically mention environmental factors

2."The results of discriminant analyses are presented in Figure 2. Confidence regions about means differed significantly (p < 0.05), indicating that genetics impacted foliar nutrient compositions more than environmental conditions. As a result, tissue compositions, that integrate the effects of soil properties, rootstock and orchard management, should be diagnosed separately for each orange variety". I do not think we can see "indicating that genetics impacted foliar nutrient compositions more than environmental conditions." from precent data. The results only indicate that we can not discriminant where the leaves come form and cannot know where the soil come from. If we can only know these two conclusions, how we follow the authors hypothesis "tissue compositions, that integrate the effects of soil properties, rootstock and orchard management, should be diagnosed separately for each orange variety"?

3.The author uses 60 tons of annual output as a standard for nutritional imbalance. But there are many reasons for the lack of yield, how can we use the data from this study to blame nutrition? Furthermore, the authors want to apply machine learning methods to present the value of the study, but only select 10 special cases as In comparison, this amount of samples only accounts for a very low proportion of all 716 samples.

Ultimately, if the authors want to show the goodness of machine learning, they should apply traditional nutritional comparisons so we can know exactly what is going on. For example, ANOVA analysis is used to compare the nutritional status of different yield areas.

Author Response

Comments and Suggestions for Authors

  1. Although the authors add a lot of sampling information, we still don't know the actual situation. Since the source of the database is unclear, and this study is based on the database for AI modeling, it is difficult to assess the quality of this research.

The authors mention in the abstract that 716 orchards were sampled, but only 25 were sampled for soil (line 117). There is no way to tell how many orchards the leaves were taken from, we only know that 100 leaves were sampled from each orchard. Which cultivar of each orchard were planted for each sampled soils and leaves? Are they sufficient to represent the so-called 716 orchard plots? Judging from the current information, only 3% (25/716) of the data sources have analyzed soils, how to confirm the author's hypothesis"" We hypothesized that varietal ionomes and "environmental factors" can be amalgamated to generate regional nutrient standards across Brazilian orange orchards using tools of machine learning and compositional data analysis. "? The authors specifically mention environmental factors

  1. We agree. Soil tests discarded as well as the DA conducted on soil test values. We assumed like Munson and Nelson (1990) that tissue tests include information on soil test, fertilization and crop management, leaving tissue tests as features to be related to fruit yield. Genetic features and tissue tests were fully documented.

2."The results of discriminant analyses are presented in Figure 2. Confidence regions about means differed significantly (p < 0.05), indicating that genetics impacted foliar nutrient compositions more than environmental conditions. As a result, tissue compositions, that integrate the effects of soil properties, rootstock and orchard management, should be diagnosed separately for each orange variety". I do not think we can see "indicating that genetics impacted foliar nutrient compositions more than environmental conditions." from precent data. The results only indicate that we can not discriminant where the leaves come form and cannot know where the soil come from. If we can only know these two conclusions, how we follow the authors hypothesis "tissue compositions, that integrate the effects of soil properties, rootstock and orchard management, should be diagnosed separately for each orange variety"?

  1. Soil tests were discarded from DA.

3.The author uses 60 tons of annual output as a standard for nutritional imbalance. But there are many reasons for the lack of yield, how can we use the data from this study to blame nutrition? Furthermore, the authors want to apply machine learning methods to present the value of the study, but only select 10 special cases as In comparison, this amount of samples only accounts for a very low proportion of all 716 samples.

  1. Growers target 60 tons in Brazil. We tested 40 and 50 tons, but both were less accurate than 60 tons (Table 2). Nevertheless, other yield targets may be tested depending on yield goal. We selected the ten nearest compositional neighbors based on the Euclidean distance for comparison with the defective specimen as suggested by Munson and Nelson (1990) as normal vs. abnormal cases. For cases less documented in the data set, less than ten high-yielding neighbors may be available. The principle is to compare successful to unsuccessful specimens and address the most limiting nutrients using grower’s success stories.

Ultimately, if the authors want to show the goodness of machine learning, they should apply traditional nutritional comparisons so we can know exactly what is going on. For example, ANOVA analysis is used to compare the nutritional status of different yield areas.

Indeed, we used discriminant analysis for comparison because nutrient compositions are intrinsically multivariate as already stated by Lagatu and Maume (Lagatu, H., and Maume, L. 1934. Le diagnostic foliaire de la pomme de terre. Ann. l’École Natl. Agron. Montpellier 22, 50–158) and Holland (Holland, D.A. 1966. The interpretation of leaf analysis. J. Hort. Sci. 41, 311-329). Both DA and KNN returned that the scion is of prime importance. Rootstock could be hardly used as variable because scion and rootstock were most often closely associated.

Academic Editor Comments

I have gone through the manuscript quickly and following issues seems to be of major concern :
i. As many 716 observations reveal no meaning , need to be split rootstock-scion combinationwise .

  1. Most often, scion and rootstock were closely associated (Table 1). This is why rootstock added little to model precision. Much more data must be collected to discriminate between scion and rootstock effects.

Orchards age of 7-15 years,  i feel , still too early to establish the peak orchard productivity , any reason to select this orchards age  .  

  1. The 7 to 15 years of age is a range required to reach a certain stability in orange production (Aragay et al., 2021?). That's why we chose this age group.

How did you identify Oxisol versus Ultisol , how do these two soil orders maintain a differential nutrient-supply-chain.??

  1. Most often, authors use ‘tropical soils’ as general statement. We were more specific using ‘Oxisols’ and ‘Ultisols’ that are the dominant soil great groups in the state of São Paulo. Those soils are known to fix phosphorus tightly (p. 15, 2nd paragraph). However, the data set could be hardly split into more groups to develop soil-specific nutrient references. Nevertheless, scion, rootstock and nutrient balances alone returned an AUC of 0.796.

How those nutrient imbalances differ in Oxisol to Ultisol through a given rootstock-scion combination. ?.

We split observations into rootstock-scion combinations in Table 1. Splitting even more the data set would be useless due to imbalanced numbers of scions and rootstocks. We assumed that, as recommended, growers tackled limiting factors in those soils such as low P levels, hardpans and soil acidity.

  1. The rootstock effect on six  scion varieties is not visible vis-a-vis  available nutrient supply .Is there any need to look at nutrient imbalances 9 e.g. Ca x B) with respect to specific rootstock -scion combination .  
  2. Several nutrient balance models could be elaborated but the results of statistical analysis remains the same due to orthonormality of the ilr variables. The idea is to reduce redundancy among components and reach D-1 degrees of freedom to tackle model overfitting. We thus used ilr variables to reduce the number of features from D nutrients to D-1 ilr values. As mentioned above, ‘Hamlin’, ‘Valência’ and ‘Pêra’ were generally associated with the same rootstock, making it practically impossible to set apart the effects of scion and rootstock. As a result, rootstock effect impacted model precision very little.

    Such studies direly need to be cross-validated through progressive nutrient response studies through rootstock-scion combination  to add a greater strength to outcomes of machine learning . No doubt , such studies do add to our better understanding about cross-talk between different nutrients , but again under a given rootstock-scion combination , the same would be more appealing .  
  3. This reviewer is right. This will be the next step to put together the experimental results and the observational data from crop surveys across Brazil to generate a large and diversified national data set through data sharing and more experimental work.

    Is it rationale to fix  cut off yield of 60 Mg ha-1 regardless of  so many rootstock -scion combination ( six rootstocks and six scions ) and two contrasting soil orders?.  .
  4. Our goal was to build an accurate ML model. Scion, rootstock and nutrient balances returned AUC of 0.796 in cross-validation where yield cutoff was fixed at 60 ton ha-1 (we used ton instead of Mg to avoid confusion with the chemical symbol of magnesium). Nevertheless lower yield goals for the nearest neighbors at local scale could be extracted from the data set for specific scion X rootstock combinations yielding less than 60 ton ha-1.

Round 3

Reviewer 3 Report

1. We still do not understand why there are only 10 results of each true negative (TN) and true positive (TB) specimens were shown. If authors' hypothesis is correct, authors could use more date to prove. For example, authors could add a Scatter plot, and put all model training set and predicted set on it. Then we can understand better.

2. Authors hope to use KNN to judge the Nutrient diagnosis, however, we cannot see the "diagnosis", because we did not know the reasons caused the lower yield. We can only know the grouping by model ( if model is well), then we put a new date to test the nutrient  condition is close to high productivity or lower productivity. And then we still do not know which nutrient elements or which range lead to the different productivity.

3. We do not know how Authors could result the order of nutrient limitations? by KNN? If from KNN we should see the whole 716 data set, if not from KNN, how authors could claim they use "integrative tools of machine learning and composition data analysis"? if KNN could not provide the suitable range for each Nutrient, how we can only know the orchards conditions is closed to high productivity or lower productivity, we cannot know which Nutrient we need to apply. If we did not know these information from KNN model, why we need to use integrative tools???

4. we cannot see the efficiency of integrative tool in this research for Nutrient diagnosis, we can only see the efficiency of evaluation for suitable or not suitable. It just a Yes/ No evaluation, not a diagnosis. I think authors could explain this opinion more clearly in the manuscript and consider to change the title to restraint in only limited in the evaluation not to diagnosis. And the results form integrative tools seem still cannot to provide a new Nutrient guidelines for orange, authors sold also notice when they do these dissertation

Author Response

Dear Reviewers and Editor

All questions raised are answered in the attached file.

The new version of the article follows below the answers.

Sincerely,

Prof. Danilo Eduardo Rozane

Round 4

Reviewer 3 Report

the present version is much better than before. However, the response to reviewer are much clearer than those description in  manuscript. Please add some more information to explain how the results could be use in the future and how it could help the orange growers in Brazil, that will be easier to know the contribution of thiw work.

Author Response

The academic editor should read the paper attentively before formulating those comments.

In response to questions:

i) As many 716 observations reveal no meaning , need to be split rootstock-scion combinationwise . Orchards age of 7-15 years,  i feel , still too early to establish the peak orchard productivity , any reason to select this orchards age  .  How did you identify Oxisol versus Ultisol , how do these two soil orders maintain a differential nutrient-supply-chain.?? .How those nutrient imbalances differ in Oxisol to Ultisol through a given rootstock-scion combination. ?

We explained how Oxisols and Ultisols may differ in terms of soil classification usable as potential features. Nutrient imbalances involve not only soil class but also nutrient management. The scion-rootstock combination is another feature that impacts tissue nutrient balances and crop yield. Now we have 551 well documented complete observations to account for soil classification, soil tests and climatic indices.

ii) The rootstock effect on six  scion varieties is not visible vis-a-vis  available nutrient supply .Is there any need to look at nutrient imbalances 9 e.g. Ca x B) with respect to specific rootstock -scion combination .  

This comment is out of context (see Tables) or meaningless. There are three scions not six in the revised version. The Ca x B interaction is already treated as clrCa and clrB.

iii. Such studies direly need to be cross-validated through progressive nutrient response studies through rootstock-scion combination  to add a greater strength to outcomes of machine learning . No doubt , such studies do add to our better understanding about cross-talk between different nutrients , but again under a given rootstock-scion combination , the same would be more appealing .  

We agree, but this was not the objective of the paper. Please address the issue formulated in the paper.

iv. Is it rationale to fix  cut off yield of 60 Mg ha-1 regardless of  so many rootstock -scion combination ( six rootstocks and six scions ) and two contrasting soil orders?

There are three scion x rootstock combinations in the revised version. We examined not only 60 but also 50 Mg ha-1. The rationale is to relate a target variable (fruit yield) to yield-impacting features.